# Structure and Undulations of Escin Adsorption Layer at Water Surface Studied by Molecular Dynamics

**DOI:** 10.3390/molecules26226856

**Published:** 2021-11-13

**Authors:** Sonya Tsibranska, Anela Ivanova, Slavka Tcholakova, Nikolai Denkov

**Affiliations:** 1Department of Chemical and Pharmaceutical Engineering, Faculty of Chemistry and Pharmacy, University of Sofia, 1164 Sofia, Bulgaria; st@lcpe.uni-sofia.bg (S.T.); sc@lcpe.uni-sofia.bg (S.T.); nd@lcpe.uni-sofia.bg (N.D.); 2Department of Physical Chemistry, Faculty of Chemistry and Pharmacy, University of Sofia, 1164 Sofia, Bulgaria

**Keywords:** escin, aescin, viscoelastic surface layers, surface undulation, molecular dynamics

## Abstract

The saponin escin, extracted from horse chestnut seeds, forms adsorption layers with high viscoelasticity and low gas permeability. Upon deformation, escin adsorption layers often feature surface wrinkles with characteristic wavelength. In previous studies, we investigated the origin of this behavior and found that the substantial surface elasticity of escin layers may be related to a specific combination of short-, medium-, and long-range attractive forces, leading to tight molecular packing in the layers. In the current study, we performed atomistic molecular dynamics simulations of 441 escin molecules in a dense adsorption layer with an area per molecule of 0.49 nm^2^. We found that the surfactant molecules are less submerged in water and adopt a more upright position when compared to the characteristics determined in our previous simulations with much smaller molecular models. The number of neighbouring molecules and their local orientation, however, remain similar in the different-size models. To maintain their preferred mutual orientation, the escin molecules segregate into well-ordered domains and spontaneously form wrinkled layers. The same specific interactions (H-bonds, dipole–dipole attraction, and intermediate strong attraction) define the complex internal structure and the undulations of the layers. The analysis of the layer properties reveals a characteristic wrinkle wavelength related to the surface lateral dimensions, in qualitative agreement with the phenomenological description of thin elastic sheets.

## 1. Introduction

It is known that certain classes of surface-active species may form highly viscoelastic adsorption layers [1,2]. Examples are some synthetic polymers [3,4], polysaccharides [5,6,7], proteins [8,9,10,11,12,13], phospholipids [14], low-molecular-mass surfactants [3,15,16], mixtures of such substances [17,18], and/or particles [19,20,21,22,23]. The rheological properties of the adsorption layers formed from such substances have been extensively studied both at the air–water and oil–water interfaces. Among the compounds forming highly viscoelastic layers are also some types of natural amphiphilic molecules called “saponins”, found in extracts from various plants. The saponin molecules are composed of a hydrophobic rigid scaffold (aglycone) and one to several hydrophilic sugar residues. Recently, a large number of articles has deepened our understanding of the various classes of saponins, their interfacial properties, and several related phenomena; see, e.g., Refs. [24,25,26,27,28,29,30,31,32].

The experimental studies showed that the triterpenoid saponins, which exhibit viscoelasticity in shear deformation at the air–water interface, have similar viscoelastic behaviour in dilatational deformation as well [28]. Interestingly, the surface elasticity of the saponins decreases rapidly with the increasing amplitude of deformation and the effect is very pronounced for tea saponin, escin, and ginseng saponins [28]. This decrease in the surface elasticity with deformation amplitude most probably reflects the breaking of the attractive intermolecular bonds and the related disruption of the tight molecular packing in the surface layers. As a general trend, the elastic moduli at the oil–water interface are substantially lower than those at the air–water interface [27]. The most probable explanation of this trend is that the oily molecules penetrate in-between the saponin molecules in the adsorption layer and, thus, decrease the magnitude of the attraction between the adsorbed saponin surfactants [27].

The rheological properties of the surfactant adsorption layers are known to significantly affect the behaviour of the bulk foams and emulsions [33,34,35]. One of the main challenges of the colloid science in general and in the studies of saponins in particular is to determine the relationship between the properties of the adsorption layers and the properties of the bulk disperse systems (foams and emulsions).

Saponins are not only natural amphiphiles (surfactants) with high surface activity—many of them exhibit high biological activity as well. Due to this unique combination, saponins are used and are potentially important in several branches of technology and science. To apply them efficiently, however, further in-depth understanding of the relationship between their molecular structure, the complex interactions within their adsorption layers, and their non-trivial surface properties [28,36] is needed. Such molecular-level understanding may be provided by molecular dynamics (MD) simulations.

The main component of chestnut (*Aesculus hippocastanum*) seed extracts is the saponin escin, or aescin, (ESC) which may be produced with very high purity [37,38]. Some of the physicochemical properties of escin are truly remarkable. Escin adsorption layers have exceptionally high elastic moduli, above 1100 mN/m. In some experimental studies, the authors observed the formation of wrinkles on the water surface which reflect the unusual viscoelastic properties of such highly elastic layers. These wrinkles also indicate very low surface tension upon surface compression, leading to spontaneous buckling of the adsorption layer. Wrinkle formation was also reported for insoluble monolayers of solid particles, lipids, and some specific proteins [16,23,24,39,40,41,42], while most saponins have high solubility in water. Some authors linked the shape, period, and amplitude of the surface wrinkles to the surface elasticity, which is also bound to the related bending mechanics of the surface layer [43,44]. In addition, escin adsorption layers have small gas permeability and relatively high surface viscosity of ca. 130 Ns/m which are important for foam stability with respect to bubble Ostwald ripening [28,36].

The origin of those peculiar properties of escin adsorption layers is not fully understood at the molecular level. In our previous study [45], we employed classical atomistic MD simulations to analyse in detail the molecular orientation and the interactions in the supramolecular structures (domains) formed in dilute escin monolayers. We revealed significant differences in the behaviour of the charged and neutral forms of the escin molecules (they contain ionizable carboxyl groups) and explained them in terms of the intermolecular interactions between the adsorbed escin molecules. The high elasticity of the neutral escin layers was attributed to the combination of several complementary attractiveforces, including one rather specific interaction, intermediate in strength between the classical hydrogen bonds and the dipole–dipole interactions, acting between the hydroxyl groups in the sugar residues in the escin molecules. The modelling in this first work, however, was made with dilute adsorption layers (low surface coverage).

In a following study [46], we carried out atomistic MD simulations for 49 escin molecules, arranged in dense adsorption layers on the water surface, at two different areas per molecule. The results showed that the molecules in the dense adsorption layers are less submerged in water and adopt a more upright position, as compared to the dilute layers. For solvation of the large hydrophilic fragments, which are lifted above the water surface in the dense saponin layers, significant number of water molecules are trapped around the hydrophilic sugar groups. To maintain the preferred orientation of the escin molecules with respect to the water surface, the denser adsorption layers acquire a significant local spontaneous curvature. The substantial elasticity of the neutral escin layers was confirmed to originate from the combination of intermolecular interactions, complemented with several new structural factors: substantial flexibility of the surfactant head-groups, rigidity of the aglycon part and significant normal displacement of the molecules with respect to the water surface to allow their tight packing in the layer, which creates the observed curvature of the interface. Long-range order throughout the entire layer was revealed in these simulations, which signifies the role of the collective behaviour of the adsorbed escin molecules, while no regular crystal lattice was observed. It has remained unclear from these previous studies whether the observed features of the modelled adsorption layers would be preserved in escin layers of larger area, with box dimensions being much larger than the area of the individual escin molecules.

Only a few other publications report results from MD simulations of saponin molecules [47,48,49,50,51,52,53,54,55,56,57,58,59,60,61,62]. These papers study the kinetics, mechanism of formation, and shape of micelles assembled in bulk saponin solutions. The structure of the single saponin molecules is also analysed in terms of their conformation stability and structure-property relationships [47,48,54,58,61]. Docking studies are performed [50,51,52,53,55,56,57,59,60,62] to understand the type and strength of interactions of saponins with different substrates, e.g., lipid membranes or bio-receptors.

MD simulations can also be used to outline the effect of the hydrophobic groups on the rheological behavior of polymer solutions [63]. Some authors expand this approach by quantifying the rheology of aqueous solutions of surfactant micelles [64] by coarse-grained non-equilibrium MD simulations. The observed shear-thickening effect is explained by the micellar breaking and dispersing of the surfactant molecules.

Several papers studied the wrinkles of thin layers and the elasticity parameters which correspond to this layer wrinkling [65,66,67,68,69,70,71,72,73]. These studies explored the morphology of buckling, the mechanical fundamentals of this phenomenon, the chemical functionalization, and the mechanical properties of the molecules involved (including graphene). Wrinkles in different soft materials were observed and theoretical interpretation of this phenomenon was proposed [67,70,73]. It was shown [73] that the bending stiffness and the in-plane stiffness are key indicators that signify the intrinsic mechanical properties of thin graphene sheets.

The above literature overview shows that there are limited MD simulations of escin adsorption layers. Extending our previous studies towards an evaluation of the parameters related to the viscoelastic behaviour of escin layers at the gas–water interface provides additional information to understand better the molecular origin of the experimentally observed phenomena. In the current work, MD simulations of relatively large dense escin adsorption layers are made and the molecular orientation, the structural features, and the intermolecular interactions are characterized and compared to those in the smaller systems studied previously [45,46]. A relationship between the molecular behaviour and the experimentally determined rheological characteristics of the escin layers is proposed and discussed.

## 2. Results and Discussion

MD simulations are carried out for a model of 441 escin molecules (Figure 1), adsorbed on the surface of a water subphase, with area per escin molecule of 0.49 nm^2^. This surface coverage (Appendix A) corresponds to the experimentally measured one in dense adsorption layers of escin [28]. The initial configuration is of homogenously (in-plane) distributed molecules. However, as in our previous study [46], the surface rapidly became corrugated along the simulations, with a characteristic undulation wavelength comparable to the size of the simulation box, see Figure 1. Note that undulations, very similar in shape and wavelength, with a period of ≈34 ± 4 nm were detected experimentally in adsorption layers of other saponins, when observing the surface of oil-in-water emulsion drops (see Figure 4 in Ref. [35]).

In the following, we characterize the structure of the formed layer, the respective intermolecular interactions, and the undulation characteristics and compare them to those determined in our previous study [46].

### 2.1. Hydrogen Bonding of Escin Molecules

Figure 2a presents the evolution of the number of hydrogen bonds within and between the escin molecules in the studied model. It is clearly seen that there is a decreasing trend of the number of H-bonds with time. This decrease is due to the fact that the number of H-bonds is influenced substantially by the intermolecular orientation. In our previous study [45], we identified that the hydrogen bonds between ESC molecules are predominantly formed between OH groups of two of the terminal sugar residues. At the same time, the aglycones orient close to each other and feature dispersion attraction. Hence, the observed trend means that the molecules reorient over time to achieve these requirements. Moreover, the escin molecules thrive to maximize the number of hydrogen bonds of their sugar residues with water (see Figure 2b) and to minimize the steric hindrance of all fragments, thus minimizing the overall free energy of the system. It is also evident from the plot that the system is (almost) relaxed after 450–500 ns and the number of H-bonds changes slightly afterwards, reaching a constant average value after ca. 700 ns. This is considered as the characteristic relaxation time for the models of this size (441 molecules, abbreviated ESC441 further on). When the relaxation time for the current model is compared with that of the smaller model (49 molecules) at the same area-per-molecule of 0.49 nm^2^ [46] (termed ESC49 further on), it can be seen that the relaxation time increases with the model size, as expected. The relaxation time of the systems (441 vs. 49 molecules) is approximately proportional to the square root of the number of molecules.

On the other hand, the number of the hydrogen bonds escin–water (Figure 2b) increases with time concurrently to the decrease of the inter-escin H-bonds. This increase can be attributed to the drive of the water molecules to penetrate and to hydrate better the hydrophilic head-groups of the escin molecules. The process continues until an equilibrium degree of hydration is reached that corresponds to ca. 15.3 H-bonds with water per escin molecule. It can be seen that the number of ESC-water H-bonds increases faster in the initial period, when compared to the ESC-ESC H-bonds decrease. In other words, the initial process of hydration progresses much faster than that of surfactant reorientation. A plateau is reached also in the ESC-water profiles after 700 ns, confirming the relaxation time for achieving the optimum H-bonding in the model. All analyses discussed below are performed only on the relaxed part of the trajectory (viz. last 300 ns).

### 2.2. Position of the Escin Molecules Relative to Water

Analysis of the mass density profiles along the z-axis (Figure 3) provides the characteristic layer thicknesses in direction normal to the interface and the relative position of the escin molecules with respect to the equimolecular dividing surface (EDS, denoted in green in Figure 3).

The peak of escin is located at the edge of the bulk water density (Figure 3), above the EDS, which confirms the pronounced surface activity of ESC. Comparison to the results at low surface coverage [45] and with smaller models at similar surface coverage [46] shows that the molecules in the large dense layer are much less submerged in water. This difference with the systems with low surface coverage is probably due to the tighter packing between the escin molecules, which enhances the total hydrophobicity of the system. Thriving to pack and to reduce the intermolecular steric repulsion, the escin molecules adopt a more upright position in the large dense layers, which leads to emerging of some of the molecular fragments above the water surface. Escin, however, contains a large hydrophilic part, which needs to remain solvated. This is effectuated by trapping water molecules around the hydrophilic sugar heads of the escin molecules and dragging these water molecules well above the EDS. This conclusion is illustrated by the long tail in the water density profile, encompassing the entire escin population along the z-axis.

This plot also evidences that the half-width of the escin peak (6.50 ± 0.10 nm) is wider than that in the smaller models [45,46] where the respective peak half-widths are as follows: 2.19 ± 0.02 nm (49ESC-70), 3.00 ± 0.02 nm (49ESC-49) and 1.08 nm for the diluted model with 16 escin molecules. This much larger thickness of the large dense model is related to the substantial amplitude of the surface undulations (Figure 1). The undulations in the large model result in a much broader population of the z-coordinates by escin molecules and, hence, in a wider density peak. In order to quantify better the orientation of the escin molecules with respect to the surface and to the neighbouring escin molecules, several additional analyses are carried out.

### 2.3. Interfacial Orientation and Intermolecular Ordering of the Escin Molecules

The tilt of the escin aglycone is used as a measure of the molecule orientation with respect to the water surface. This tilt is quantified as the angle closed between the z-axis of the simulation box (Figure 1) and a vector spanning the whole aglycone, as shown in Figure 4a. This analysis is done for the last 300 ns of the trajectories and the results are displayed in Figure 4b. The plot shows that the tilt of the ESC molecules varies in the range from 120° to 160°. This result means that the molecules are usually not perpendicular to the surface, while they are much more upright oriented in the condensed layer than in the diluted models [45], where the value of 110° was preferred. The observed range of tilt angles is similar to the one we observed in the smaller model of dense adsorption layer, ESC49. The distribution of the values is as follows: 90° ÷ 120° ≈ 16 %, 120° ÷ 150° ≈ 47 % and 150° ÷ 180° ≈ 37 %. The larger tilt (> 120°) is predominant, which means that escin is oriented primarily at angles closer to 180°, viz. closer to perpendicular alignment of the aglycone at the interface. The smallest populated angle is 90° which corresponds to parallel orientation with respect to the surface. However, angles between 90° and 100° are very rare (< 2 %). The difference between the dense and the diluted models [45] is worth commenting on. In the models with lower surface coverage, the most populated tilts were in the range 110° – 114°, while in the models with high surface coverage the angles are much larger: 159° to 168° for models with 49 molecules and in the range 120°–150° in the current study.

The angles discussed above are defined with respect to the fixed z-axis. Since the layer becomes undulated, part of the measured tilt may originate from the curvature of the undulated layer, as shown in the ESC49 model [46]. Therefore, we also estimated the actual tilt of the escin molecules, relative to the local normal to the undulated layer. For this purpose, we fit the quasilinear part of the surface to a straight line (for details see ref. [46]) by plotting the z-coordinate of atom C19 (this atom is selected as a reference because it is at the base of the rigid aglycone, see Figure 4a) of all ESC molecules as a function of their y-coordinates. From the fit, we determined an average slope of the surface relative to the z-axis of 37 ± 1°. The average slope in the current model is practically the same as that for ESC49. Correcting the average surfactant tilt with this value results in a reduction of the ESC angle with respect to the actual (undulated) water surface down to 105 ± 20°. The latter angle is very close to the tilt registered in the small models [45].

This last result confirms the proposed mechanism (based on the models with 49 molecules) for reducing the steric repulsion between the escin molecules: the adsorption layer bends and acquires an undulated shape upon compression to preserve the preferred mutual orientation of the escin molecules. Hence, it can be summarized that a tilt close to 110°–120° is the most favourable orientation of ESC, i.e., each molecule best accommodates the neighbouring molecules and can make an optimal number of hydrogen bonds with the neighbouring water molecules.

To characterize the intermolecular orientation in such a multimolecular system, it is usually accepted to use different order parameters [74]. In the current study, we employed the rotational order parameter (OP) for the last 300 ns of the trajectory.
(1)$$OP=\left(\frac{1}{N}\right)\frac{1}{\tau }\sum_{i}^{N(molecules)}\sum_{i}^{\tau (Time)}\cos(180-\alpha )$$

The definition of the order parameter (OP) via Equation (1) requires taking a cosine of the difference between 180° and the angle between the aglycone and the z axis (α). This OP can be averaged over time (τ), over molecules (N), or over both. Order parameter values close to 0 designate a fully disordered system, typical for isotropic liquids, whereas magnitudes close to 1 are characteristic of a crystalline structure with a high degree of order in the crystal lattice.

The order parameter for ESC441 as a function of time is shown in Figure 5a. There, we block-averaged the OP over all molecules for trajectory parts of 50 ns. It can be seen that the OP decreases slightly (even though the adjacent averages do not differ within the standard deviations) up to 850 ns and after that it remains constant. This result indicates that fine rearrangement of the molecules in the adsorption layer occurs up to 850 ns. A comparison of Figure 2 and Figure 5a shows that the coarse rearrangement of molecules takes place for ca. 700 ns, while the fine additional alignment continues up to 850 ns. This follow-up reorientation is probably associated with a small rotation of the molecules without significant displacement and change in the hydrogen bonding between the neighbouring molecules. From Figure 5a, it is also seen that the averaged OP in the relaxed part of the trajectory is around 0.735, which corresponds to a structure with a relatively high degree of order. However, the value is significantly lower than that for regular crystalline arrangement.

Figure 5b contains the order parameter averaged over the last 50 ns of the trajectory for each ESC molecule. It may be seen that the different molecules have very different OP values, varying from close to 0 to almost 1. This graph shows that there is no uniform crystalline arrangement of the escin molecules along the entire surface. Instead, there are domains with higher degrees of order (high OP) followed by regions with low degrees of order (low OP). This result confirms the hypothesis of a surface domain structure that was made in the previous study of ESC49 [46] and which was proposed by Golemanov et al. [25] based on their experimental data.

The variance in the averaged OPs could be attributed to the nonidentical molecular orientations in the different parts of the undulation wave, to the formation of domains or to combination of both. The molecules with high OP are surrounded by other molecules with the same orientation. This leads to optimal hydrogen bonding and strong attraction between the neighbouring molecules. Such molecules form the core of the domains or are located in the steep part of the wave. At the periphery of the domains or in the extrema (maxima or minima) of the wave, however, the molecules exhibit a loss of order because the neighbouring escin molecules have a different orientation. There, the molecules do not have optimal hydrogen bonding and are characterized with higher energy.

Next, we decided to check for the presence of long-range order. We calculated the radial distribution functions (RDFs) of the distance between the reference C-atoms from the aglycones (shown in green in Figure 6b). Both 2D RDFs (in the *xy* plane) and 3D RDFs were generated.

It is clearly seen that the curves for the 2D RDF (Figure 6a) and 3D RDF (Figure 6b) differ significantly from those typical of a fluid surface but also do not resemble those of crystals with regular lattice. At the same time, there are multiple peaks up to a distance of several nanometers, especially well visible in the 2D RDF. In other words, there is a partial long-range order in the escin layers, while no true crystal lattice is formed.

The highest peak of the 2D RDFs is deformed by a shoulder at the smaller distances and splits into two at the side of the larger distances. In the 3D RDF, this peak also splits into two separate maxima at around 1 nm. These graphs reflect the systematic displacement of the first neighbours along z, as discussed above.

When comparing these results with those obtained in our previous study [46], it may be concluded that as the size of the model increases, the distance up to which surfactants order also increases. This shows that there is a long-range order in the escin layers, at least up to several nanometers of lateral distance.

### 2.4. Surface Undulations and Curvature

Here, we analyse the wave-like shape of the surface in terms of its amplitude and period. With this aim in view, we determined the *x*, *y*, and *z* coordinates of the reference atom C19 (shown in green in Figure 6b) for all molecules and for the entire trajectory of 1000 ns. Then, we drew these coordinates as 3D plots (Figure 7a) for each trajectory of 50 ns. In these plots, the values of the *x*, *y* and *z* positions are averaged over time. We determined the shape of the surface thereof. We found that the shape of the layer resembled a wave in the *y*-*z* plane. We then plotted *z* as a function of *y*, time-averaged for each 50 ns trajectory period (Figure 7b). Next, we smoothed the data to a middle line describing the positions (Figure 7c) and fitted these smooth data to a sine function (Figure 7d).

Several types of functions were tested for the final fit and the best description was obtained using the following sine function:(2)$$z\left(y\right)=y_{0}+A\sin\left(\frac{2\pi y}{B}+C\right)\;$$

In Equation (2), A is the amplitude, B is the period, C is the horizontal phase shift and *y*_0_ is the vertical phase shift. Figure 8 shows the results for the amplitude, period and phase shift, as a function of time, obtained after fitting all 20 trajectory periods of 50 ns. It may be seen that the amplitude (Figure 8a) varies from 2.07 nm to 2.14 nm with standard error of 0.04 nm without clear dependence, while in the relaxed part of the trajectory (after 850 ns) the respective value is constant ≈ 2.1 nm. Figure 8b collects the results for the period of the wave. It increases slightly with time up to 800 ns and then remains at ≈ 16.6 nm. The horizontal phase shift (Figure 8c) decreases with time to 900 ns and remains constant for the last 100 ns ≈ −0.5 whereas the vertical phase shift (Figure 8d) goes through a maximum at ca. 500 ns and then levels off at ≈ 8.0 nm.

We compared the obtained data for the wave period with the size of the box, which is 14.7 nm in *x* and *y* directions. It can be seen that the period predicted by the fit is a bit larger than the box size. The ratio between the predicted period and the size of the box in the *y* direction (a fit was made along this direction) is 1.14, which shows that the wave spans the entire periodic box. We also did this type of fit for the last 50 ns of the trajectory of the ESC49 models [46]. We found that for the model with 0.70 nm^2^ area per molecule the period is 7.5 nm and for the one with 0.49 nm^2^ area per molecule it is 4.5 nm. The ratio between the predicted period and the size of the box for ESC49 models is 0.89 for the model with 0.70 nm^2^ area per molecule and 0.92 for the one with 0.49 nm^2^ area per molecule. All these values are close to 1, which means that the wave spans the entire periodic box for all models studied so far. The amplitudes for both models were also determined to be 0.45 nm at 0.70 nm^2^ and 1.03 nm for 0.49 nm^2^ area-per-molecule, respectively.

The latter observation suggests that either our models are too small for the formed corresponding surface wave or that the waves are able to adjust their period to the lateral dimensions of the interface. We can compare the two models with the same area per molecule of 0.49 nm^2^ whereby the same initial structure is used, i.e., the model with 441 molecules is three times bigger along the x and y axis compared to ESC49. So, it may be expected for the large model to have at least three periods if the wave is maintained as in the model with 49 molecules. However, no such behaviour is observed. The period increases with increasing box size, regardless of the initial structure. This result of the simulations supports the second hypothesis and is most likely related to a decrease of the bending energy in the larger box by enabling a longer period and larger amplitude of the wave.

In order to compare the obtained waves in the different models, we plot in Figure 9 the predicted z-coordinates as a function of x- or y-coordinates after fitting.

The big difference between the waves for the different models is evident. In a smaller box (pink curve in Figure 9) the period is smaller, but the amplitude is larger. This leads to sharp minima and maxima on the surface, which are associated with high bending energy. In a larger box (blue curve in Figure 9) but with the same number of molecules, the period is larger but the amplitude is smaller, which leads to less bending energy. As the box increases together with the number of molecules, an even longer period occurs. The behavior of the layer resembles a spring, which extends to the maximum possible size (which is the box size in this case) which serves as a mechanism for layer relaxation.

As a final step, we decided to check which characteristics of the layer can be predicted from the available experimental data and to compare them qualitatively with the properties of the simulated layers. For this purpose, we used the methodology described in the work of Erni et al. [44], where it is suggested that the physics of wrinkle formation in thin elastic layers may be used as a sensitive indicator of interfacial rheology. The general expression proposed in [44] for the wrinkle wavelength is:(3)$$\lambda =2\surd \pi {\left(\frac{D}{\sigma }\right)}^{\frac{1}{4}}L^{\frac{1}{2}}$$
where *D* is the flexural rigidity of the elastic sheet, σ is the surface tension, and *L* is the length of the sheet. One sees that the wavelength λ depends on the lateral dimension of the layer, so the considerations made above for the escin layers with different box sizes are in agreement with this model. To link the wavelength λ to the interfacial elastic properties, the authors proposed the following equation for the flexure rigidity of an elastic sheet of thickness *h*:(4)$$D=\frac{Yh^{3}}{\left(1-\nu^{2}\right)}$$
In this equation, ν is the Poisson’s ratio and *Y* is the pseudo-bulk value of the Young’s modulus of the layer material. The latter can be expressed approximately as:(5)$$Y=\frac{Y_{s}}{h}$$
where *Y*s is the (2D) Young’s modulus and can be calculated from the equation:(6)$$Y_{s}=2G_{S}^{\prime }\left(1+\nu \right)$$
In this equation, $G_{S}^{\prime }$ is the interfacial shear modulus and the Poisson’s ratio can be determined from the following expression:(7)$$\nu =\frac{G_{D}^{\prime }}{2G_{S}^{\prime }}-1$$
where $G_{D}^{\prime }$ is the interfacial dilatational modulus.

Therefore, from known values of $G_{D}^{\prime }$, $G_{S}^{\prime }$, and *h* one could determine ν, *Y*s, *Y*, and *D* and compare them with data known from the literature. To make these estimates, we used some of the parameters determined experimentally in Refs. [28,36,75] – the interfacial shear modulus $G_{S}^{\prime }=0.057\;N/m$, interfacial dilatational modulus $G_{D}^{\prime }=0.165\;N/m$, and layer thickness *h* = 2.6 nm. From these values and using Equations (4)–(7) we calculated ν ≈ 0.447, *Y*s ≈ 0.165 N/m, *Y* ≈ 6×10^7^ Pa and *D* ≈ 1.4 ×10^-18^ J. The obtained magnitude of the pseudo-bulk Young’s modulus *Y* corresponds to the values typical for waxy materials [76]. The flexural rigidity of the adsorption layer is relatively high (≈ 340 *k*_B_*T*) when compared to the rigidity of typical lipid bilayers and is similar to values observed for some polymers [77]. Note that the estimated values of ν, *Y*, and *D* are very reasonable from a physical viewpoint. These estimates support the idea that the undulations observed in the molecular dynamics simulations are in qualitative agreement with the elasticity parameters evaluated from the experimental data of escin adsorption layers.

## 3. Molecular Models and Computational Protocol

The force field AMBER99 [78,79] is used in combination with the TIP3P [80] water model. The initial geometry and the MM parameterization of escin are described in our previous study [45]. The model system (Figure 1) is built by translating a one-molecule unit cell in the *xy* plane on the nodes of a regular square lattice. Then, the hydrophilic parts of the escin molecules are hydrated by 43,452 water molecules. The simulations are carried out in the presence of 10 mM NaCl, which requires 9 Na^+^ and 9 Cl^-^ in the model. The respective sizes of the periodic box are: *x* = 14.70 nm, *y* = 14.70 nm, *z* = 17.18 nm. The model system is simulated in PBC with 5 nm of vacuum introduced on both sides along the z-axis (which is perpendicular to the interface) to truncate the periodicity in this direction. All MD calculations are at constant temperature of 293 K, maintained by a Berendsen thermostat [81], i.e., in NVT ensemble. These conditions and the respective concentrations are selected to correspond to those in the experimental measurements [28,75].

The potential for the van der Waals interactions is Lennard-Jones and is truncated at a distance of 1.2 nm with a switching function activated at 1 nm. The electrostatic interactions are calculated with PME [82,83,84] where the direct summation is truncated at 1.2 nm with a switching function turned on at 1 nm. The time step is 2 fs and the equations of motion are integrated with leap-frog. The lengths of all hydrogen-containing bonds in escin are fixed with LINCS [85], and those in water with SETTLE [86]. The energy of the model system is minimized first. Then, the molecules are heated to the desired temperature and equilibrated following a standard procedure [45,46]. The total energy and the temperature are checked for attained equilibrium. The overall period of initial equilibration is 1 ns. Then, a production trajectory with length of 1000 ns is generated for the model system. Frames are written down every 1 ps. The last 300,000 structures are subject to statistical analysis. The respective sections of the trajectories are selected for the analyses because there the systems are already stationary in terms of the target property. Hydrogen bonds are used as the most sensitive structural parameter to verify when equilibrium is reached (Figure 2). The density profiles are obtained by block averaging of 50 ns trajectory segments to assess the statistical accuracy.

The simulations are carried out with the program package Gromacs 5.1.2 [87]. VMD [88] is used for visualization of the trajectories.

## 4. Conclusions

Atomistic 1 μs long molecular dynamics simulations of large (203139 atoms), dense adsorption layers of escin, adsorbed at the vacuum–water interface, are performed. Key interfacial structural characteristics of the system are analysed and compared to those for smaller models of escin layers [45,46]. It is found that the larger size of the model induces diminished surfactants submergence in water and the adoption of a more upright position relative to the surface. The local orientation after accounting for the local surface curvature and slope, however, is close to the tilt observed in the smaller models [45,46].

The orientation, number of nearest neighbouring molecules, and the area per molecule are not affected by the size of the model. Neither is the long-range intermolecular order, which encompasses the entire layer and is maintained by the same specific intermolecular interactions as in the smaller models.

To sustain the preferred orientation with respect to the water surface, escin molecules align in well-ordered domains and the layer acquires a significant spontaneous curvature, forming undulating waves in one of the lateral dimensions. It turns out that the main characteristics of the wave-like surface pattern, i.e., period and amplitude, depend on the size of the simulation box. The layer shape can adjust to the size of the box in a way, which allows efficient relaxation of the interfacial stress. This spring-type relaxation mechanism is in line with the phenomenological model [44] enabling the estimation of two viscoelastic parameters of the adsorption layers, the Young’s modulus and the flexural rigidity, from outcome of experimental surface rheology measurements. The simulation data obtained with dense escin layers provide additional information concerning the peculiar properties of these exceptionally viscoelastic layers.

## Figures and Tables

**Figure 1 molecules-26-06856-f001:**
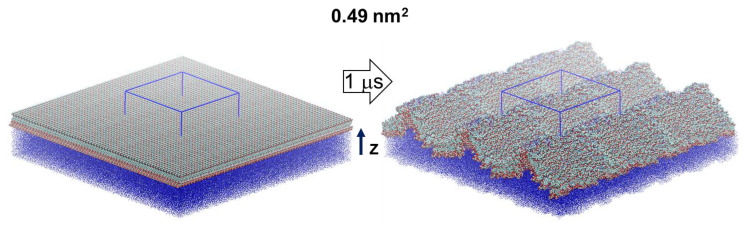
Simulated molecular models of 441 escin molecules (replicated along the lateral dimensions) at the vacuum/water interface with 0.49 nm^2^ area per molecule. The initial configurations (left) are shown together with a final snapshot from the MD simulations (right); the periodic box is outlined in blue; the thick arrow denotes the length of the simulation and the thin black one—the direction of the z-axis of the coordinate system.

**Figure 2 molecules-26-06856-f002:**
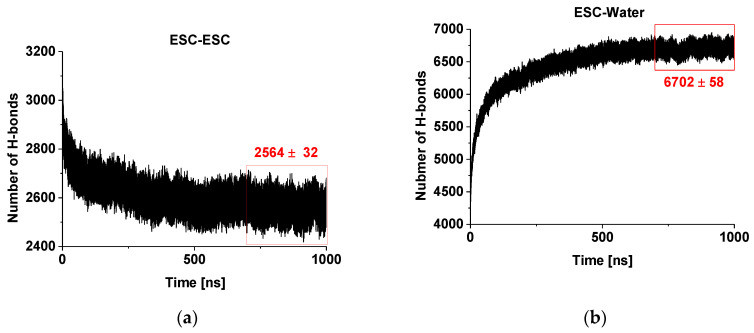
Number of hydrogen bonds in and between escin molecules (**a**) and between escin and water molecules (**b**) as a function of time.

**Figure 3 molecules-26-06856-f003:**
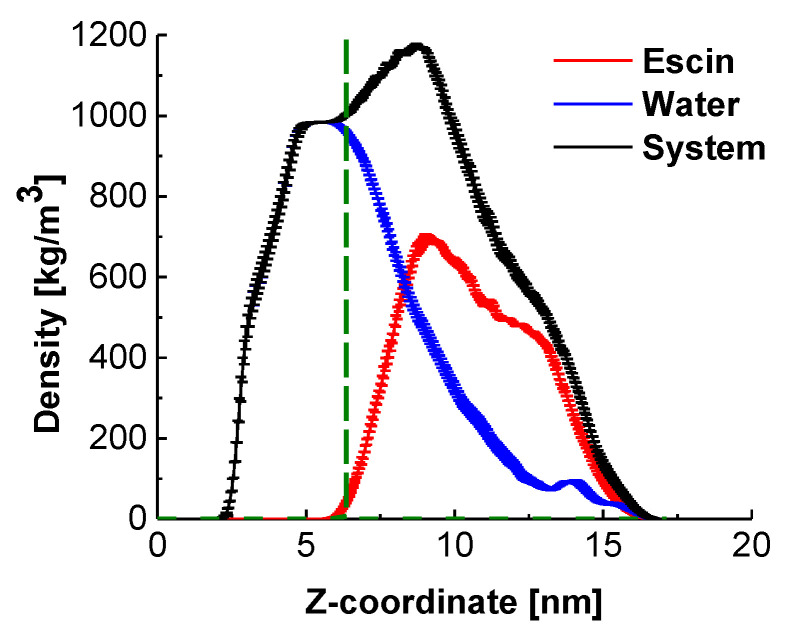
Density profiles in direction normal to the interface for the entire system, for water, and for the escin molecules in the model; the fuzzy ranges denote the standard deviation. The term “system” encompasses all components (water, escin molecules and electrolyte ions).

**Figure 4 molecules-26-06856-f004:**
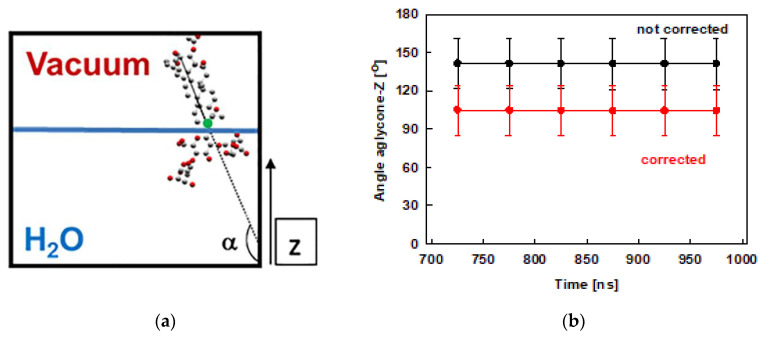
Schematic representation of the angle closed between the aglycone of escin and the z axis (the reference aglycone C19 carbon atom is marked by green circle) (**a**) and the angle evolution during the MD simulations of ESC441 (**b**). The angles are block-averaged over all molecules for trajectory parts of 50 ns.

**Figure 5 molecules-26-06856-f005:**
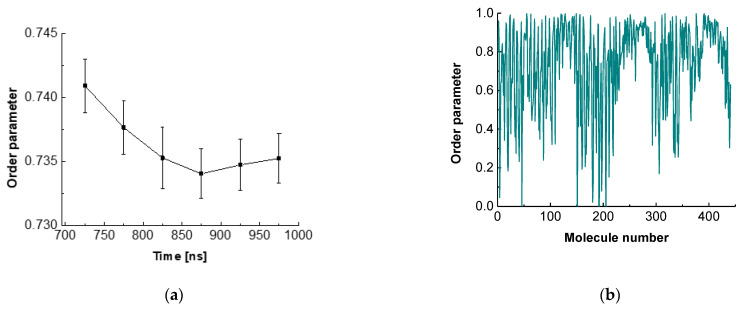
Order parameter of escin as a function of time (**a**) or of molecule number (**b**).

**Figure 6 molecules-26-06856-f006:**
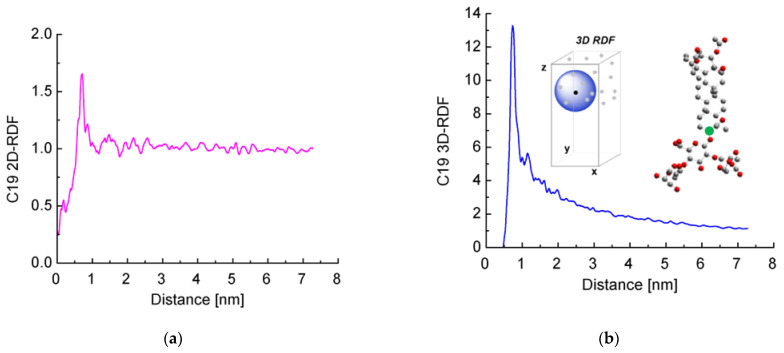
Radial distribution functions of the distance between a reference aglycone carbon atom (marked with a green circle in the inset) of escin molecules calculated in the plane of the layer (**a**) or along the three dimensions of space (**b**).

**Figure 7 molecules-26-06856-f007:**
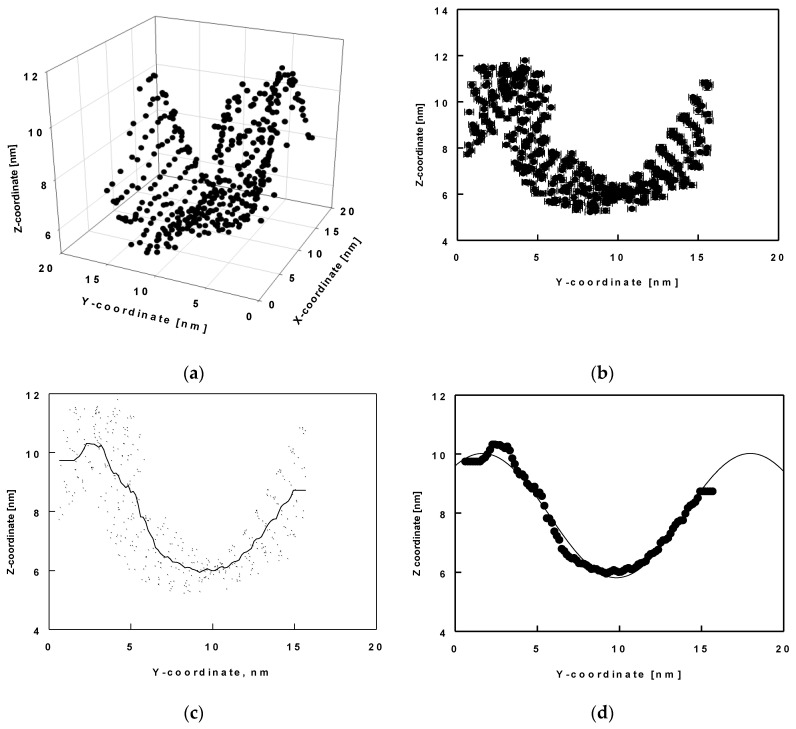
Average *z*-coordinates as a function of the average *x* and/or *y*-coordinate of the atom C19 of all ESC molecules in the model: three-dimensional plot of *x*, *y* and *z* coordinates (**a**); *z*-coordinates as a function of the average *y*-coordinates (**b**); interpolation of all C19 coordinate to one smooth curve (**c**); fit of the smooth curve with a sine function (**d**); error bars denote standard deviations from averaging over 50 ns trajectory parts.

**Figure 8 molecules-26-06856-f008:**
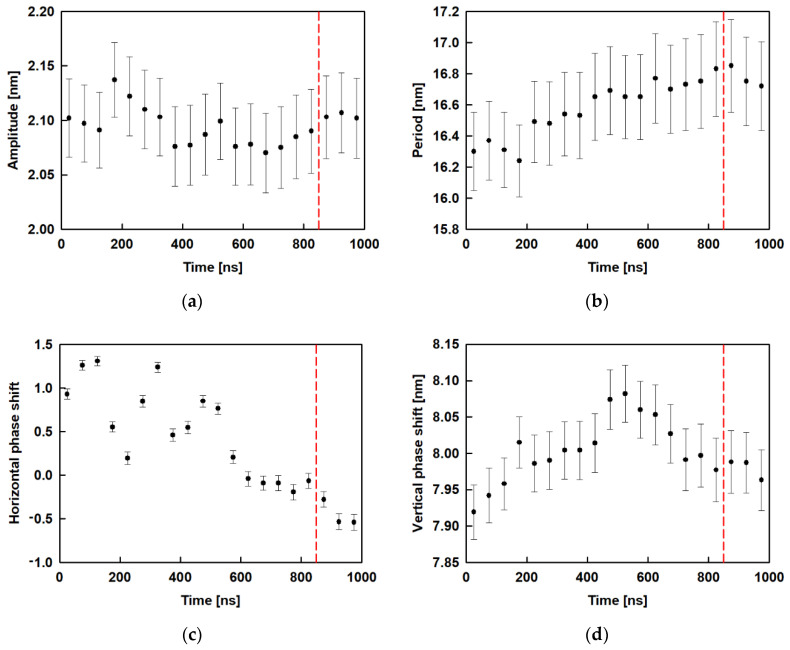
Evolution of the average characteristics of the surface curvature (obtained after fitting with a sine function): amplitude (**a**); period (**b**); horizontal phase shift (**c**); vertical phase shift (**d**); the red dashed line is drawn at 850 ns and denotes the beginning of the relaxed trajectory part.

**Figure 9 molecules-26-06856-f009:**
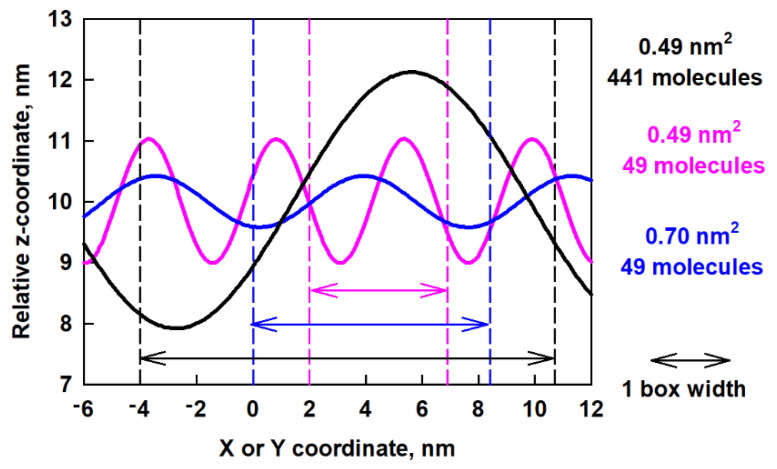
Predicted z-coordinates as a function of x or y-coordinates after fitting of the surface curvature of the ESC49 and ESC441 models; the arrows of each color denote one full box width of the respective model.

## Data Availability

Not applicable.

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
