# Peer review of "Structure and Undulations of Escin Adsorption Layer at Water Surface Studied by Molecular Dynamics"

_molecules, 2021, doi:10.3390/molecules26226856_

Round 1

Reviewer 1 Report

The present submitted draft by Tsibranska et al. describes in clear words the molecular dynamics simulation of a large area of escin molecules at an computational air/water interface. The study tries to shed light on the unusual buckoing and wrinkling behaviour found of such adsorption layers in experiments. The molecular dynamics simulation performed and described in detail clearly demonstrates the unusual behaviour and explains the interplay between H-bonds, dipol and strong interaction to cause this behaviour due to the unusual structural composition of saponin molecules. Overall the presentation of the work done, the analysis of the computational results and the drawn conclusions are confident. Overall there is no distinct drawback in the presenttaion of the paper in it can be accepted for publication as submitted.

Author Response

We are grateful to the Reviewer for the appreciation of our work.

Reviewer 2 Report

This manuscript describes structure and undulations of escin adsorption layer at water surface. MD simulations are used to investigate the characteristics of relatively large dense escin adsorption layers, and a relationship between the molecular behavior and the experimentally determined rheological characteristics of the escin layer is proposed and discussed. However, this work needs significant revision before accepting by this journal.

  1. “This decrease is due to the fact that the number of H-bonds is influenced substantially by the intermolecular orientation” (page 4, line 161). What are the specific influencing factors? Atoms, groups or other factors?
  2. The half-width of the escin peak in Fig. 3 is about 7 nm, which may have error.
  3. The equations on page 6 line 228 make me confused. What are the meanings of the equations and the parameters in the equations?
  4. The explanation of the angle aglycone-z correction is not very sufficient (page 6 line 238-248). There is no fitting straight line mentioned in line 242 in Fig. 6. What does the ‘C19’ represent (page 6 line 243)? The explanation of ‘C19’ should be given when it first appeared.
  5. The author should give more intuitive data in the comparison with previous work, rather than reading these articles at the same time, if they want readers to understand their work easier.

Author Response

The responses are attached as a separate file.

Round 2

Reviewer 2 Report

My comments were fully addressed in this revision with reasonable answers and edits. I am satisfied with this revision and am happy to recommend the acceptance of this work in the current form.